# Surface Modification of Ti6Al4V ELI Titanium Alloy by Poly(ethylene-alt-maleic anhydride) and Risedronate Sodium

**DOI:** 10.3390/ma16155404

**Published:** 2023-08-01

**Authors:** Joanna Szczuka, Mariusz Sandomierski, Adam Voelkel, Karol Grochalski, Tomasz Buchwald

**Affiliations:** 1Institute of Materials Research and Quantum Engineering, Poznan University of Technology, 60-965 Poznan, Poland; tomasz.buchwald@put.poznan.pl; 2Institute of Chemical Technology and Engineering, Poznan University of Technology, 60-965 Poznan, Poland; mariusz.sandomierski@put.poznan.pl (M.S.); adam.voelkel@put.poznan.pl (A.V.); 3Faculty of Mechanical Engineering, Poznan University of Technology, 60-965 Poznan, Poland; karol.grochalski@put.poznan.pl

**Keywords:** endoprosthesis, titanium alloy, polymer, silanization, risedronate sodium, surface modification

## Abstract

With the simultaneous increase in the number of endoprostheses being performed, advances in the field of biomaterials are becoming apparent—whereby the materials and technologies used to construct implants clearly improve the implants’ quality and, ultimately, the life of the patient after surgery. The aim of this study was to modify the titanium alloy Ti6Al4V ELI used in the construction of hip joint endoprostheses. This is why the continuous development of biomaterials is so important. This paper presents the results of research for a new application of polymer poly(ethylene-alt-maleic anhydride) as a drug release layer, placed on the surface of a titanium alloy. The obtained layers were analyzed using Raman spectroscopy (spectra and maps), Fourier transform infrared spectroscopy (spectra and maps), contact angle measurements as well as scanning electron microscopy and energy dispersive spectroscopy imaging and topography analysis. The results confirmed that the polymer layer obtained on the plate surface after the alkali heat treatment process is much better—it binds much more polymer and thus the applied drug. In addition, a longer and more gradual release of the drug was observed for the alkali heat treatment modification than for H_2_O_2_ solution.

## 1. Introduction

The development of technology and medicine makes it possible to increasingly successfully replace human tissues and organs with their synthetic counterparts. This is the case, for example, in arthroplasty—people after injuries or as a result of disease can replace a damaged joint or bone fragment, which allows them to return to a "normal" life. In order for a given implant to take effect, it is important to select appropriate materials—not only of high strength (hip or knee endoprosthesis), but also that are biocompatible. One such material is the Ti6Al4V ELI titanium alloy, which belongs to the two-phase α+β alloys. It is characterized by its high susceptibility to heat treatment, which is important when creating endoprosthesis elements tailored to the patient’s needs. This alloy is obtained by introducing an appropriate amount of elements stabilizing the α and β phases. Aluminum (Al) is an element that stabilizes α↔β transformations, reduces the density of alloys, and increases the thermal stability of the β phase. On the other hand, vanadium (V) is an isomorphic element, stabilizing the β phase, increasing its plasticity, and lowering the temperature of the α↔β allotropic transformation. Titanium alloys are biologically inert and non-magnetic materials. They are most often used as a building material for artificial sockets and metaphyseal pins [1,2,3]. Our test material was titanium alloy Ti6Al4V ELI; it is a certified material (based on the ASTM F136 standard [4]) used in the construction of endoprostheses.

Among the most commonly used medications supporting the treatment of osteoporosis as well as supporting convalescence after arthroplasty surgery are those containing zoledronate and risedronate. According to the data of the Polish National Health Fund, the reimbursement amount for these drugs in 2017–2020 was PLN 32,540,188.74 (the value of the drugs in general was PLN 38,613,887.64 for 682,801 packages) [5], while in the provision of hip and knee arthroplasty in the years 2014–2020, PLN 8 billion was spent on drugs [6,7].

Therefore, it was decided to modify the surface of the titanium alloy with a drug; however, between the alloy and the drug, it was decided to use an intermediate layer—a polymer, variants of which are widely used in the construction of materials for medical applications. Polymers are used to create dental composites, bone cements, or elements of endoprostheses [8,9]. They also play an important role in the process of drug transport—they form capsule shells or drug-doped membranes [10,11]. They are characterized by their biodegradability, depending on the type of polymer, when placed in the body, a given layer disintegrates after a period of several hours to even several weeks. PEAMA is a popular compound for the improvement of the mechanical and physicochemical properties of plastics [12]. This polymer is most often found as a copolymer in the literature, but in this work it was decided to check its direct influence—its independent participation in the reaction. It is significantly cheaper than the currently popular PGAs. For example, considering Sigma Aldrich prices, per 1g: PEAMA—EUR 6.89; PGA—EUR 185. This can significantly affect the final cost of such a modified material.

Due to the structure of the polymer and the drug, it was decided to use 1,1′-Carbonyldiimidazole (CDI)—it is one of the most popular compounds used in the coupling process [13]. Its presence is necessary for the coordination of the PEAMA layer with Zn ions. The formation of imidazolium rings on the PEAMA surface is not only expected to coordinate zinc ions, but also has great potential to increase biocompatibility and osseointegration. This is due to the fact that such layers have previously supported bone growth (e.g., ZIF-8 layer) [14]. In the next and last step of the modification, the drug was attached by a coordination bond (between the Zn and RSD molecules) [15]. The use of zinc ions for the drug sorption process was based on the results of studies on zinc zeolites [16], where the obtained Zeo-Zn-RSD coating was characterized by good strength properties as well as the parameters (amount) of the drug released from the surface.

The aim of the research was to develop a method of modifying the titanium alloy to obtain a surface capable of drug release. The modification was carried out in two ways: for the surface of the plates modified with the alkali heat treatment method, and for the tiles after immersion in H_2_O_2_. The key layer was a PEAMA polymer deposited on a titanium alloy after the silanization process. In the next step, a drug from the bisphosphonate group was attached in the conjugation reaction. The final and most important step was the measurement and quantitative analysis of the sorption and desorption of the drug from the surface of the modified plates. Scanning electron microscopy, energy dispersive spectroscopy, Raman spectroscopy, Fourier transform infrared spectroscopy measurement of the surface contact angle, UV-Vis spectroscopy and measurement of topography were used to analyze the surface and the obtained layers.

A schematic diagram of the stages of sample modification is presented in Figure 1.

## 2. Materials and Methods

### 2.1. Preparation of the Titanium Alloy Plate

For the tests, titanium alloy Ti-6Al-4V ELI plates (from WOLFTEN Sp. z o.o., Wrocław, Poland) were used measuring 12.5 mm × 10 mm × 1 mm, percentage alloy composition: Fe—0.12; V—3.85; Al—6.15; C—0.008; O—0.13; N—0.004; Y—<0.0004; Ti—89.74.

The plates were degreased by immersing them in a solvent EM 404 (Emag) in an ultrasonic cleaner. Then, the wet grinding process was performed with sandpaper with a grammage, respectively, of 600, 800, 1000, 1500, 2000, ending with 2500. They were finally washed with water and a tetrahydrofuran (THF) solution. The samples prepared in this way were the starting point for further modifications.

### 2.2. Alkali Heat Treatment

The process proposed (first) by Kim [8] and modified by Fujibayashi [17] and Su [18] creates pores on the surface of the alloy, thanks to which a better penetration of HA is enabled, or the overall improvement in the bone-implant osseointegration. A total of 8 g NaOH (40 g/mol, Chempur, Poland) was divided into 4 parts and added one by one to the MiliQ water (20 ml). The beaker with MiliQ had been placed earlier in a mix of water and ice. After all the pellets had dissolved, the solution was poured into a new beaker with Ti-alloy plates. Next was the placement of the solution with the samples into an oven for 24 h at a temperature of 60 °C. The samples after an alkali treatment process were washed two times for 15 min in demineralized water. In the final step, the plates were put into a ceramic crucible and placed in an oven at a temperature of 600 °C for 3 h (with step 5 °C /min (sum 2 h) + 3h in 600 °C), with the plates hereafter denoted as Ti_AHT.

### 2.3. Hydrogen Peroxide (H_2_O_2_) Treatment

Hydrogen peroxide (H_2_O_2_) is used as an oxidant and as a disinfectant with antibacterial effects. When etching titanium alloys, it can create small cracks in the surface [19,20]. To modify the surface, selected plates were immersed in 35 mL of H_2_O_2_, then washed with deionized water and dried at 80 °C for 1 h. The plates prepared in this way (hereafter denoted as Ti_HO) were further analyzed.

### 2.4. Silanization Surfaces by (3-Aminopropyl)-triethoxysilane (APTES)

First of all, the silanization process consisted of preparing the solution—1.6 mL of APTES (99%, Sigma Aldrich, St. Louis, MO, USA) and 78.4 mL of methanol were dissolved for this purpose. Then, after mixing, the plates were immersed in the solution for 2 h. After this time, the plates were removed from the solution and rinsed 2 times in methanol (15 min each) to remove excess/unreacted APTES from the surface.

### 2.5. Poly(ethylene-alt-maleic anhydride) (PEAMA) Coating

After the silanization process, the plates were placed in a PEAMA (aw Mw 100,000–500,000; Sigma Aldrich, USA) solution (2.8 g of polymer dissolved in 80 mL of acetone) for 24 h while maintaining the temperature of 60 °C. The plates were then washed with acetone to remove excess polymer from the surface.

### 2.6. 1,1-Carbonyldiimidazole (CDI) Coupling and Complexing

First, the plates were placed in CDI (Sigma Aldrich, Wuxi, China)–acetone solution (1.34 g in 6.5 mL) for 24 h, maintaining room temperature, and at the end washed to remove unreacted CDI. Then, the complexation process was carried out three times in 0.5 mol Zn solution for 24 h each. After this time, the samples were rinsed three times with water in order to remove excess free ions from the surface (3 times for 15 min).

### 2.7. Risedronate Sodium (RSD) Sorption

Modified plates were placed in Eppendorf tubes and then flooded with the RSD solution (certified reference material; Sigma Aldrich, China) at a concentration of 0.05 mg/mL. The plates were left in the solution for 24 h.

### 2.8. Samples Description

Table 1 contains a brief and full description of the modifications of titanium plates. In the further part of the work, mostly abbreviated nomenclature was used to improve the clarity of the text.

### 2.9. Scanning Electron Microscopy, Energy Dispersive Spectroscopy

Scanning electron microscopy (SEM) can be successfully used to analyze biological materials [21]—it is possible to precisely check the distribution of elements on the tested, modified surface. Images were performed using an FEI Quanta 250 FEG microscope operated in vacuum mode at 10 Pa using an accelerating voltage of 10 kV, for the parameters horizontal field width (HFW) 41.4 μm and magnification 10,000× (TESCAN, Brno, Czech Republic). Using the energy dispersive spectroscopy (Bruker, Ettlingen, Germany) [22,23], the chemical composition of the sample was mapped and the EDS spectra were made.

### 2.10. Raman Spectroscopy

Raman spectroscopy is commonly used for surface analysis, e.g., biomaterials, polymers, or nanomaterials [24,25]. Two studies of the modified titanium alloy plates were carried out on the InVia^TM^ confocal Raman microscope with laser of 514.5 nm, power of 20 mW, accumulation time of 10 s, and a magnification of 50×. The laser beam was automatically focused on the plate to eliminate the influence of surface irregularities on the intensity of the Raman bands. The laser power was constantly controlled. The 1800 L/mm (514.5 nm laser) diffraction gratings were used. The all band parameter was obtained by fitting the convolution of the Gaussian and Lorentzian function. Raman spectra were acquired in the spectral range from 220 to 3500 cm^−1^. The spectra were not normalized. The changes in spectroscopic parameters resulting from the surface modification process were illustrated by a Raman maps-measurement area of 100 Ă—100 μm^2^ with steps of 10 μm. The surface analysis (Raman maps) focused on the TiO_2_ layer which was to show the differences between the pre-modification of the surface (AHT vs. HO). For the remaining samples, the Raman spectra were analyzed to determine the presence of a given compound after modification; thus, changes in the intensity of the bands could be better observed. The intensity of the characteristic band without background, which was cut off in the same way as for individual spectra, was taken for the purposes of the analysis. The spectra, from which the maps were created, were not normalized. Graphical processing of the Raman maps was performed with the Origin software.

### 2.11. Fourier-Transform Infrared Spectroscopy

Kazarian and Chan in their work [26] point out that FT-IR is one of the best methods for analyzing materials for medical applications, including biomaterials and pharmaceuticals. FT-IR spectroscopy is used for the identification and quantitative analysis of chemical compounds or their mixtures, and the determination of physicochemical properties, e.g., the determination of molecular structure and its transformation by reaction, reaction kinetics, or intramolecular dynamics. Samples can be in any state of aggregation.

Titanium alloy surfaces were scanned with an infrared spectrometer (LUMOS II, Bruker Optics, Ettlingen, Germany) under reflection mode with an analyzed surface of 900 μm × 900 μm, with step 100 μm. The spectral resolution is 4 cm^−1^, and 60 scans are acquired on each measurement point. The spectrometer is equipped with a TE-MCT detector and gold mirror as a reference. External reflection was used as the acquisition mode.

### 2.12. Contact Angle Measurement

The measurement of the contact angle of the surface of a biomaterial is now one of the first and simplest methods used to determine the properties of a material [27]. The contact angle measurement process consisted of dropping a drop of distilled water—2.5 μL—from a constant height onto the surface of the material. The drop photo was then captured with a camera at 5 s and 30 s after application. The angle between the contact surface of the drop-material and the tangent to the drop was made with the ImageJ program. For each sample, 5 drop applications were made, angle values were collected on both sides of the drop, and all results were averaged, the uncertainties of the measurement results were determined on the basis of the standard deviation.

### 2.13. UV-VIS Spectroscopy

UV-visible spectrophotometry is widely used as a drug release analysis tool [28,29]. In our research, the UV-2600 (Shimadzu, Kyoto, Japan) was applied for the determination of the risedronate concentration during the sorption and release process. Measurements were made in the range of 240–300 nm (λ max = 262 nm).

### 2.14. Structure Surface Analysis

The surface structure analysis was performed using the contact method, using a Hommel T8000 (Hommel-Tester; Hommelwerke GmbH, Waltrop, Germany) measuring device with a TKU 300 measuring head and a measuring tip with a diamond cone angle of 60 degrees. The measurement parameters were as follows: the Z axis (vertical) range ± 80 μm, the measured area 4.8 × 4 mm (of which 4.8 mm is the measurement section). Gauss filtering at 0.8 μm, 480 profiles were measured. The sampling density in the direction of the X axis (i.e., the direction of recording a single profile) was 48,001 points (interval). The measurement conditions were 20 °C ± 0.5. The analysis was performed for modifications of the Ti_HO, Ti_AHT and Ti_HO_APTES_PEAMA_CDI_Zn_RSD, Ti_AHT_APTES_PEAMA_CDI_Zn_RSD plates

## 3. Results and Discussion

### 3.1. Scanning Electron Microscopy and Energy Dispersive Spectroscopy

SEM images were taken for eight types of sample modification, which are summarized in Figure 2. Already after the first stage of modification of the surface of the titanium alloy, we can notice clear differences, where after the action of H_2_O_2_ single, loosely spaced etched areas were obtained (a) and the surface still has traces of grinding, while after the AHT the process leaves clear, densely spaced porous gaps on the surface (e). After the silanization process and the addition of PEAMA, the gaps are filled, which can be seen in both cases (b) and (f). In the case of the HO group sample, the surface of the titanium alloy still shows through in the SEM images—traces of surface grinding are visible, which may indicate poor coverage by the polymer. A similar situation can be observed at the next stages, after CDI and the covalently attached Zn, in the case of the HO modification (c), where we can see clearly distinguishable structures from the surface of the plate, which disappear in the next stage of modification (d). The situation is different for the plate from the AHT group, which was completely covered, where no etching gaps are visible (g), and in the further stage (after RSD sorption) of aggregation, the surface takes on a spongy structure (h). For all steps, we can clearly distinguish the surface of pure titanium alloy from the first step (a), which may mean that the applied layers are very thin (monolayers) or the coverage/modification is not as extensive as in the case of the AHT group plates.

The EDS analysis consisted of measuring and determining the weight share of individual elements on the surface of the modified samples. For each sample and for each element (oxygen (O), zinc (Zn), phosphorus (P)), five measurements were made, which were then averaged, with the results expressed as a percentage, marking the standard deviation of the sample. The obtained data are presented in the charts in Figure 3 (green dashed vertical line separates the samples into two groups). Comparing both groups of samples—HO and AHT—it can be seen that the latter is characterized by a higher share of oxygen on the surface than in the case of the HO samples. These data correlate with the SEM images where clear, densely spaced pores could be seen after the AHT process, which means that more -OH groups were obtained than after the H_2_O_2_ modification. A clear difference can be seen in the weight distribution of zinc (Zn). For the HO sample after RSD sorption, there is a significant increase in the occurrence of Zn than there was after the independent complexation process. For the AHT sample, a slight decrease in the occurrence of Zn on the surface was observed after the RSD sorption process. These results translate into the data obtained for the last tested element—phosphorus. A more oxidized surface had a positive effect on the subsequent stages of modification, which ultimately resulted in several times higher sorption of the phosphorus (the element is present in the structure of the RSD drug) than in the case of the group of HO samples. In addition, as a supplement to the above measurements, EDS imaging was performed, and the results are summarized in Figure 4. Both of them indicate an even distribution of the individual analyzed elements. Higher density of the element (obtaining a brighter, more intense color of the entire image surface) can be seen for the oxygen and phosphorus ((h) sample), which additionally confirms the previously discussed results regarding the weight fraction of elements.

### 3.2. Raman Spectroscopy Analysis

The primary modification in the research is the etching of the titanium alloy surface—for plates after the AHT (alkali heat treatment) process, a larger oxide layer is expected. In order to verify these differences, the Raman spectra were measured, which are then summarized in Figure 5A. It can be seen that there is a clear difference in the intensity of the bands—for Ti_AHT, they are distinct and soaring. In the next step, the surface coverage (Figure 5B) of the oxides was checked and Raman maps were made. The presented maps represent (the results of which are shown in Figure 5C) the value of the area under the plot to the baseline for the spectral range of 200–1000 cm^−1^, measurement area of 100 ÷ 100 μm^2^ with steps of 10 μm. The maps confirmed that the AHT plates are covered with more titanium oxides than the HO samples. This is important information, because, depending on the amount of -OH groups obtained on the surface, further modification steps will be required. However, it is worth noting that on the plates after HO modification there are also oxide groups, but six times less, making it difficult to notice changes in the spectrum itself.

Raman spectra were also measured for the remaining modifications, which are summarized in Figure 6. In graph 6.c, the area enlarged and separated in the sub-graph is marked in yellow; additionally, in graph 6.d, the peaks for each tested material that were used for further analysis are marked with a dashed vertical line. In Figure 6a, all modifications for the HO group are summarized, and the same results for AHT are presented (Figure 6b). PEAMA and RSD bands (both substances in powder form) are shown in Figure 6c. For a clearer comparison of the results, the spectra for the Ti_HO_APTES_PEAMA_CDI_Zn_RSD and Ti_AHT_APTES_PEAMA_CDI_Zn_RSD samples with RSD and PEAMA are juxtaposed together, on a separate graph, for the range of 900–3200 cm^−1^. In Figure 6d, the characteristic peaks of raw substrates occurring on the modified titanium alloy surface are indicated by colored dashed lines. These are, respectively, for: RSD—1059 cm^−1^, 1450 cm^−1^, 2966.3 cm^−1^, 3010 cm^−1^ (green); PEAMA—1450 cm^−1^, 2901 cm^−1,^ 2942 cm^−1^ (pink). For both modifications, peak characteristic of v(C-H)—2942 cm^−1^ and v(=(C-H))—3010 cm^−1^ bonds are clearly visible, indicating the presence of both polymer and drug on the surface. In the case of the bands at 1450 cm^−1^—v(CC) is in both cases shifted by several cm^−1^ on the modified surface. Comparing the data obtained with the SEM results, it can be concluded that similar results were obtained for both groups of modifications—as evidenced by the overlap in the occurrence of characteristic bands. Unfortunately, Raman mapping was not possible due to heating of the modified surfaces, which led to false readings of the intensity of a given band.

### 3.3. Fourier Transform Infrared Spectroscopy

Surface mapping measurements were performed for six modifications—after etching in H_2_O_2_ and AHT, the silanization process, PEAMA layer formation, surface ionization with CDI, and after drug addition. Figure 7 compiles the spectra for each tested sample and marks the area on the basis of which the area under the graph was calculated and the maps were created. The band characteristics for a given compound were selected (range marked in the boxes), the presence of which does not coincide with others: for PEAMA, the analyzed range was 1815 cm^−1^–1890 cm^−1^; after Zn adsorption of the surface 1360 cm^−1^–1480 cm^−1^; after attaching the RSD 850 cm^−1^–1165 cm^−1^. Figure 8 shows the surface maps for (a) Ti_HO_APTES_PEAMA; (b) Ti_HO_APTES_PEAMA_CDI_Zn; (c) Ti_HO_APTES_PEAMA_CDI_Zn_RSD; (d) Ti_AHT_APTES_PEAMA; (e) Ti_AHT_APTES_PEAMA_CDI_Zn; (f) Ti_AHT_APTES_PEAMA_CDI_Zn_RSD. For each modification, a legend is shown for the value of the area under the plot to the baseline (as for Raman measurements). It is clearly visible that the samples subjected to the AHT process in the first step are characterized by a higher field value and thus the intensity of the bands. It can be assumed that in the final stage, after the addition of the RSD drug, there is much more of it than in the case of the surface etched in H_2_O_2_.

Analyzing the results obtained for the surface mapping, it can be seen that the samples are not uniformly covered; however, the spread between the lowest and highest values ranges from a few (for APTES_PEAMA; Figure 8a,d) to a dozen (APTES_PEAMA_CDI_Zn; Figure 8b,e) square units. The largest discrepancy can be seen for the plate Ti_AHT_APTES_PEAMA_CDI_Zn_RSD (Figure 8f)—this difference is more than a few tens of square units, but still the measured surface shows a higher intensity than after the H_2_O_2_ process (Figure 8c).

### 3.4. Contact Angle Measurement

By juxtaposing the results of the contact angle measurements (Figure 9; samples divided into two groups by a green dashed line), the first significant difference can be observed after the formation of the -OH groups; the plates after immersion in the H_2_O_2_ solution are characterized by hydrophobicity, while the titanium alloy after AHT acquires fully hydrophilic properties. For each successive modification, changes in the values of the angles are observed, but they are opposite—for the Ti_HO, each successive modification decreases the wetting angle, where for the same modifications at AHT the opposite results (increasing hydrophobicity) were observed. It was also observed that only after the drug sorption step—RSD—the surface has the same character, hydrophilic, for both groups. The contact angle results for Ti_AHT are similar to those obtained during the 600 °C heating [30]. In contrast, the Ti_HO plate is distinguished by being more clearly and more intensely hydrophobic than it was in the case of the plate without etching. Due to various ways of preparing the surface for the tests and conducting the H_2_O_2_ etching process (various concentrations, with or without temperature) it is difficult to refer to the results of the available literature, but one thing is certain—mostly the plate prepared in this way is characterized by a contact angle above 20° [31,32].

### 3.5. UV-VIS Spectroscopy

RSD sorption measurements for the modification of the titanium alloy H_2_O_2_ and AHT were performed on the 1st, 2nd, and 7th day. The RSD solution was with a concentration of 0.05 mL/l. The results are shown in Figure 10. The modified AHT sample absorbed much more drug—five times more (70 µm) than the modification with H_2_O_2_ (14 µm). In addition, for AHT, a gradual release of the drug from the surface was observed after 24 h, almost 60% (43.7 µm) was released, whereas for H_2_O_2_, all the drug was released from the surface after 9 h.

This is very important information because it allows us to assume that the surface of the endoprosthesis modified in this way could release the drug into the patient’s body gradually—several dozen hours after the operation. In addition, based on the UV-Vis results, we can assume that the surface preparation method—AHT—despite the fact that it is more resource- and time-consuming, has a more positive effect on the final effect than the more commonly used method of etching the surface by immersion in an H_2_O_2_ solution.

### 3.6. Structure Surface Analysis

Topographic analysis was performed for samples with initial modification and for those from the last stage of modification. Data for the selected parameters, i.e., Rt—total height of roughness profile [um], Rq—root mean square deviation of roughness profile, and Sa—arithmetical mean surface roughness, are presented in Table 2, while topographic maps with Abbott curves are presented in Figure 11.

Instead of the commonly used Ra parameter, it was decided to present the Rt parameter because it is more sensitive to individual vertices and depressions.

In the surface analysis performed, the Ra parameter would give poor information about the measured profile, and its interpretation for practical use would be difficult and would not give information about the shape of the profile. This parameter is also insensitive to whether the profile has vertices or depressions (it gives an absolute value), which is particularly important in determining the possibility of polymer film formation on the surface subjected to modification. The presentation of the Rt parameter more correctly presents the functional characteristics of the surface.

The parameter Rq is statistically equal to the standard deviation of the ordinates of the profile Z(x), and individual high elevations and depressions of the profile affect its value more than the value of Ra, which in the conducted analysis of the surface texture is more correct and more faithfully reflects the nature of the irregularities of the profile.

The last parameter presented numerically is the arithmetic mean surface elevation Sa, that is, the arithmetic mean deviation of the surface from the mean surface, which is the arithmetic mean of the absolute values of the deviations of the surface elevation from the mean surface. This parameter, with reference to the spatial analysis of surface asperities, describes well its general character, corresponding to the Abbott curve.

Analyzing the obtained data, it can be seen that in both cases of etching, the surface is relatively uniformly modified (correct course of the curves), however, based on the roughness values, we can see that it is the sample after AHT modification that has a higher surface roughness (which is most visible in SEM images). Another quite interesting observation is the change in the roughness profile after the last modification, where in the case of a sample after etching in H_2_O_2_, its roughness profile is flattened because active substances penetrate into the large pores of the surface. This is also reflected in the Abbott curve (Figure 11) The opposite situation was observed for the AHT modification, where an increase in the amplitude roughness parameters was observed.

## 4. Conclusions

The aim of the study was to obtain a layer capable of sorption and subsequent desorption of the drug risedronate. The study was performed in two preliminary modifications—with titanium alloy plates etched in H2O2 (HO), and a second group subjected to alkali heat treatment (AHT).

Measurements of the Raman spectra and maps showed a significant difference in the intensity of the titanium oxides in favor of AHT, and thus we can conclude that there are significantly more of them than after immersion in H_2_O_2_. The next modification step was a silanisation process using APTES, followed by the addition of a polymer, PEAMA. Based on the Raman data, it can be concluded that in both modifications the polymer was attached to the surface; however, the FTIR results clearly show that the polymer layer for the Ti_AHT_APTES_PEAMA sample is more intense—there is much more polymer than in the case of Ti_HO_APTES_PEAMA. In addition, changes in the surface structure were confirmed by SEM images. For Ti_AHT, it is more porous, which may further promote the deposition of the layer (larger contact area).

As can be seen in Figure 11, modification of the surface by applying coatings significantly affects the nature of the material share curve and the distribution of individual amplitude values of surface irregularities. These results reflect the earlier stages of the surface analysis of the modified samples—the increased surface roughness of the AHT plates (compared to HO) had a positive effect on the formation of a polymer layer and finally drug sorption. However, it should be borne in mind that the recommended value of Sa for should be in the range of 1–1.5 µm [33]; therefore, it will be required to expand the research in the future to check other AHT parameters in order to obtain better results.

Looking at the collected results, it can be further assumed that the layers obtained on the Ti_HO plates are inhomogeneous (SEM images and contact angle results). The UV-Vis drug release studies also confirmed the presence of the drug on the surface for both groups of platelets, but it was in the case of AHT modification that much more was attached. A final conclusion can also be made that PEAMA works well as a layer for drug accumulation (occurring both for the HO and AHT groups); therefore, further research on the use of this polymer will be carried out (it is planned to create new modifications with active substances).

It is worth remembering that this is one of the first studies using this polymer for surface drug-release applications. At the moment, further research is underway on further modifications—doping with antibacterial and analgesic drugs. This will allow to check further possible applications of PEAMA. It is also planned to supplement the research with antibacterial tests to be as sure as possible about the safety of using this material in living organisms. In addition, it should be remembered that the tests were carried out on a relatively small surface of the tiles (10 mm × 10 mm × 1 mm) without any surface curvatures, therefore it is planned to conduct research on a larger scale, which will determine whether the current methodology will also be beneficial for other surface shapes—especially surface preparation by AHT.

## Figures and Tables

**Figure 1 materials-16-05404-f001:**
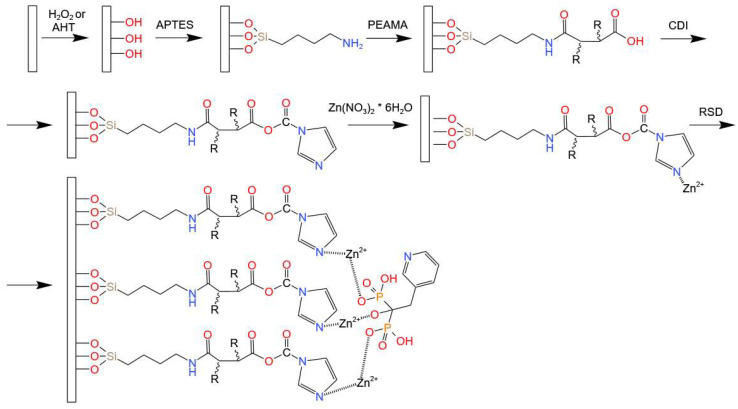
Schematic diagram of the stages of sample modification.

**Figure 2 materials-16-05404-f002:**
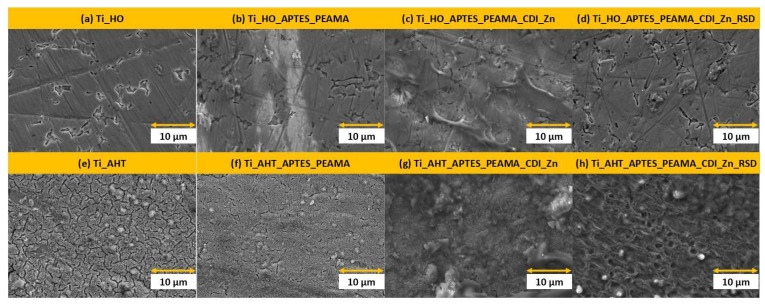
SEM images, HFW 41.4 μm, mag. 10,000×—for samples: (**a**) Ti_HO; (**b**) Ti_HO_APTES_PEAMA; (**c**) Ti_HO_APTES_PEAMA_CDI_Zn; (**d**) Ti_HO_APTES_PEAMA_CDI_Zn_RSD; (**e**) Ti_AHT; (**f**) Ti_AHT_APTES_PEAMA; (**g**) Ti_AHT_APTES_PEAMA_CDI_Zn; (**h**) Ti_AHT_APTES_PEAMA_CDI_Zn_RSD.

**Figure 3 materials-16-05404-f003:**
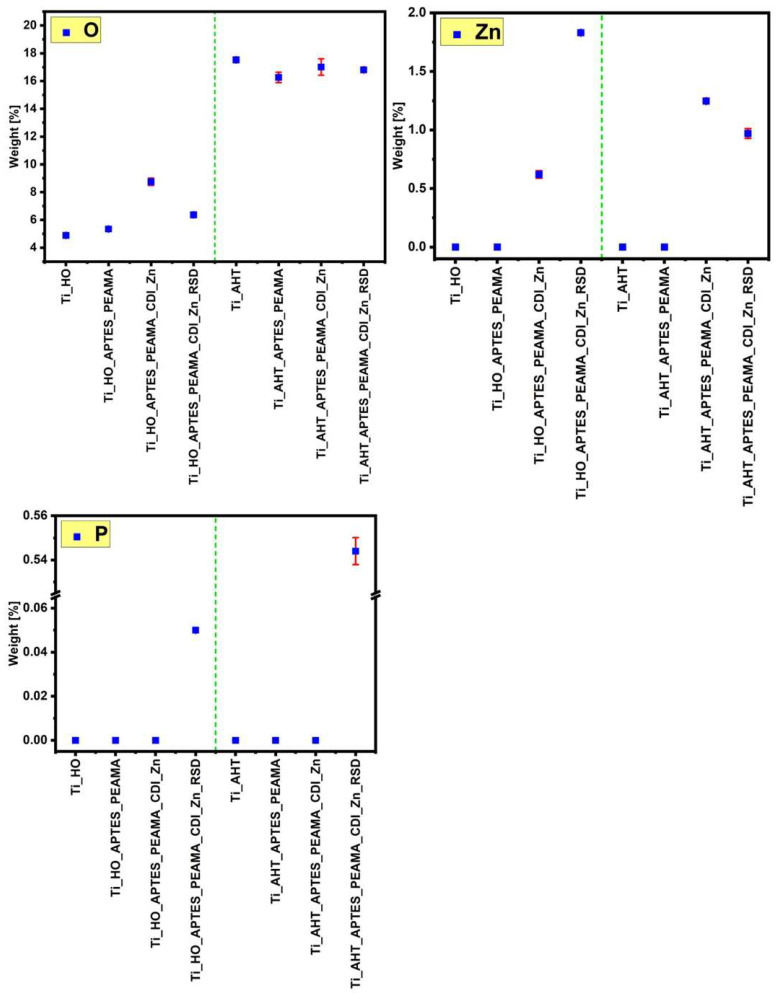
EDS analysis—the elemental weight percent values.

**Figure 4 materials-16-05404-f004:**
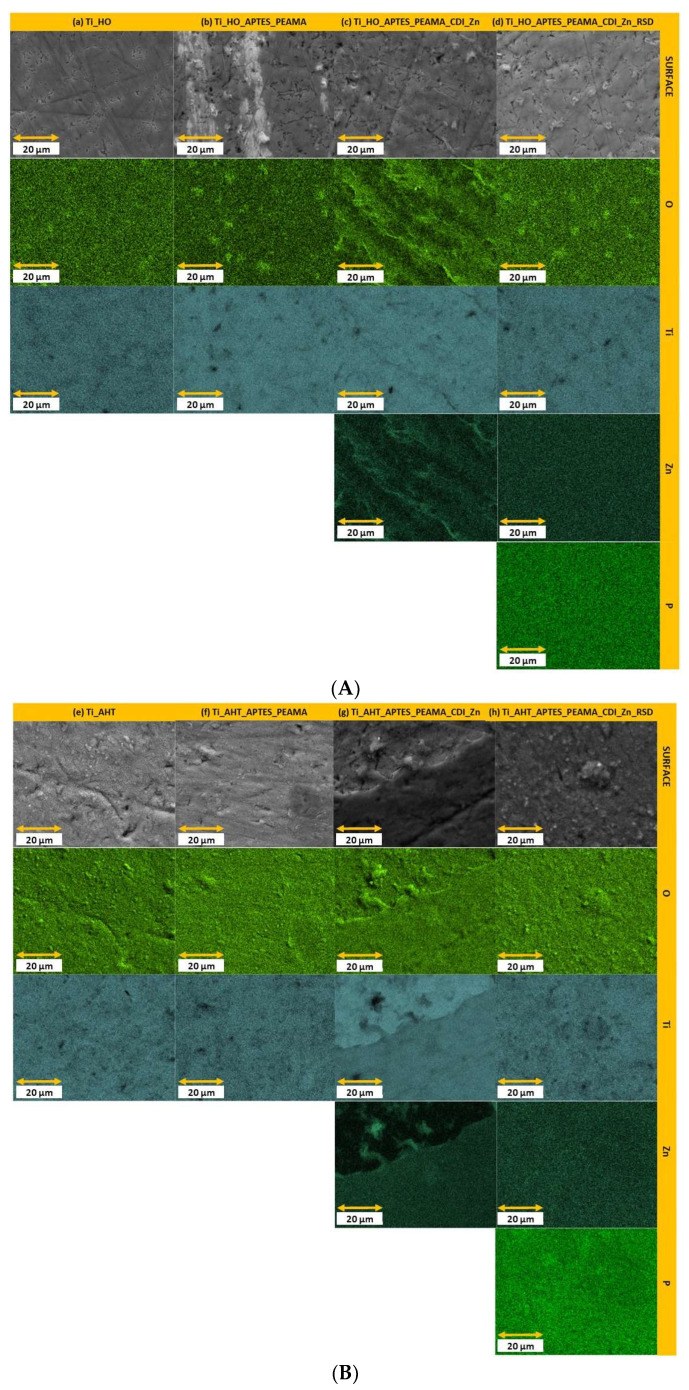
(**A**) EDS analysis (part I)—the images of elements distribution for samples: (**a**) Ti_HO; (**b**) Ti_HO_APTES_PEAMA; (**c**) Ti_HO_APTES_PEAMA_CDI_Zn; (**d**) Ti_HO_APTES_PEAMA_CDI_Zn_RSD. (**B**) EDS analysis (part II)—the images of elements distribution for samples: (**e**) Ti_AHT; (**f**) Ti_AHT_APTES_PEAMA; (**g**) Ti_AHT_APTES_PEAMA_CDI_Zn; (**h**) Ti_AHT_APTES_PEAMA_CDI_Zn_RSD.

**Figure 5 materials-16-05404-f005:**
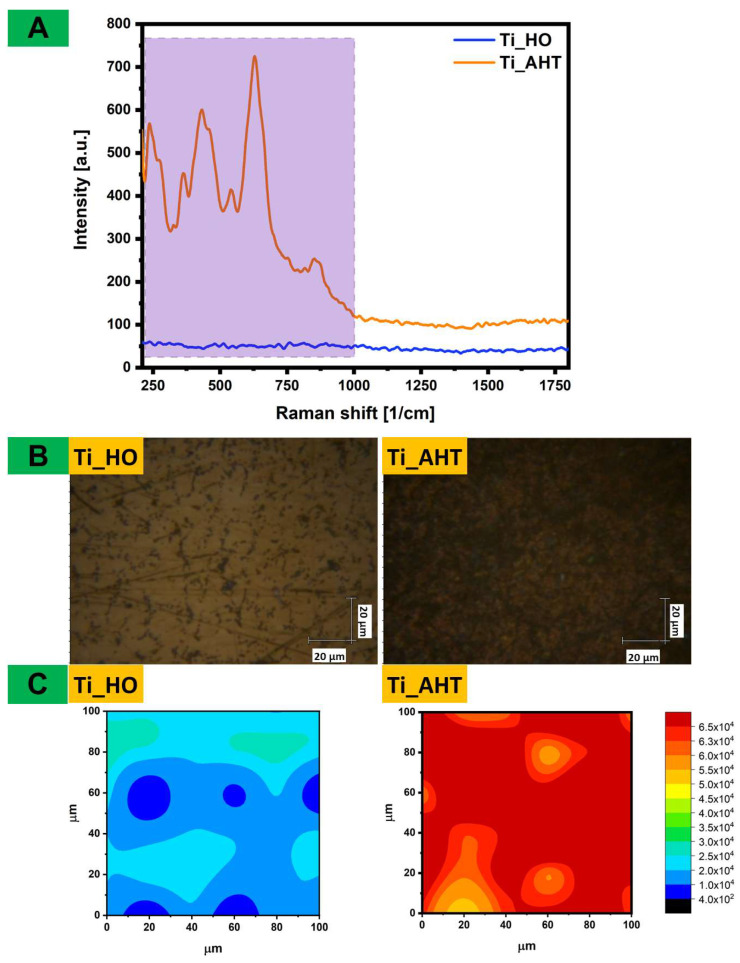
Summary of measurement for Ti_HO and Ti_AHT: (**A**)—Raman spectra, purple square presents the range of Raman shift for Raman maps (200–1000 cm^−1^); (**B**)—the area of the mapped sample surface; (**C**)—Raman maps of the surfaces.

**Figure 6 materials-16-05404-f006:**
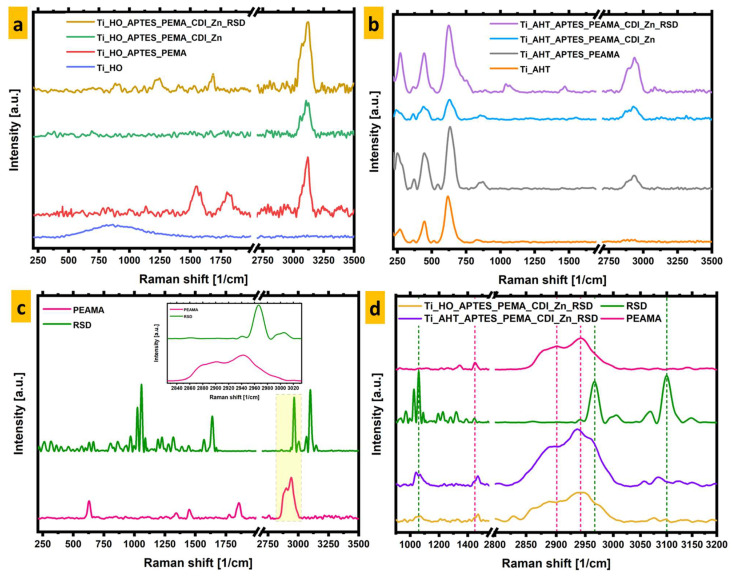
Summary of Raman spectra for (**a**)—H_2_O_2_ pre-modified group; (**b**)—group after AHT process; (**c**)—RSD (powder) and PEAMA (powder); (**d**)—comparison of spectra with final modification with reference spectra for RSD (powder) and PEAMA (powder) for the range of 900–3200 cm^−1^.

**Figure 7 materials-16-05404-f007:**
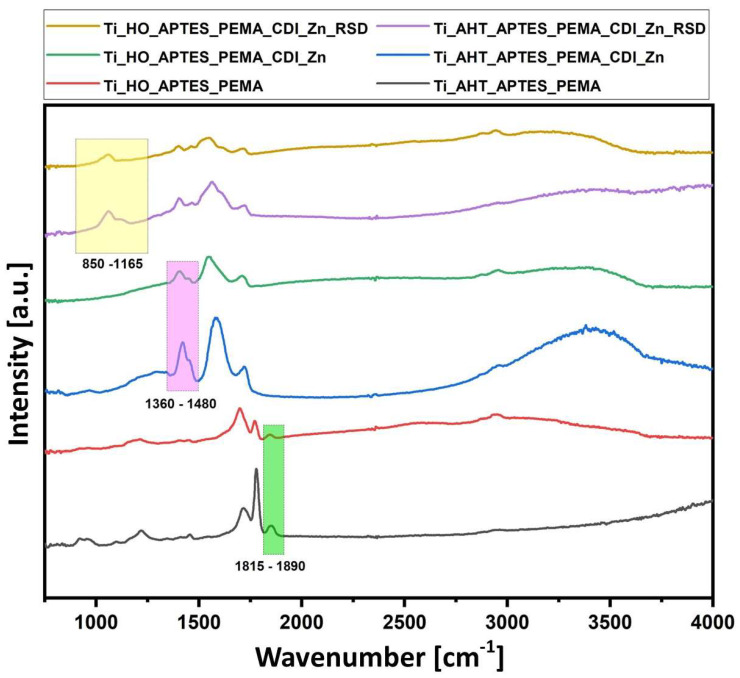
FTIR spectra for measurement samples: Ti_HO_APTES_PEAMA; Ti_HO_APTES_PEAMA_CDI_Zn; Ti_HO_APTES_PEAMA_CDI_Zn_RSD; Ti_AHT_APTES_PEAMA; Ti_AHT_APTES_PEAMA_CDI_Zn; Ti_AHT_APTES_PEAMA_CDI_Zn_RSD.

**Figure 8 materials-16-05404-f008:**
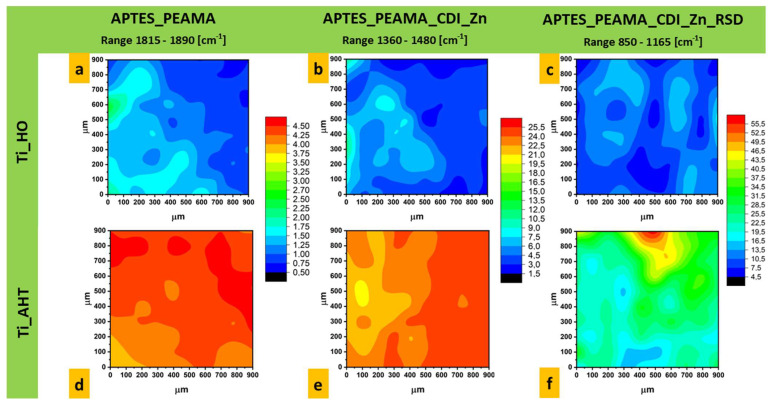
FTIR maps for samples (**a**) Ti_HO_APTES_PEAMA; (**b**) Ti_HO_APTES_PEAMA_CDI_Zn; (**c**) Ti_HO_APTES_PEAMA_CDI_Zn_RSD; (**d**) Ti_AHT_APTES_PEAMA; (**e**) Ti_AHT_APTES_PEAMA_CDI_Zn; (**f**) Ti_AHT_APTES_PEAMA_CDI_Zn_RSD.

**Figure 9 materials-16-05404-f009:**
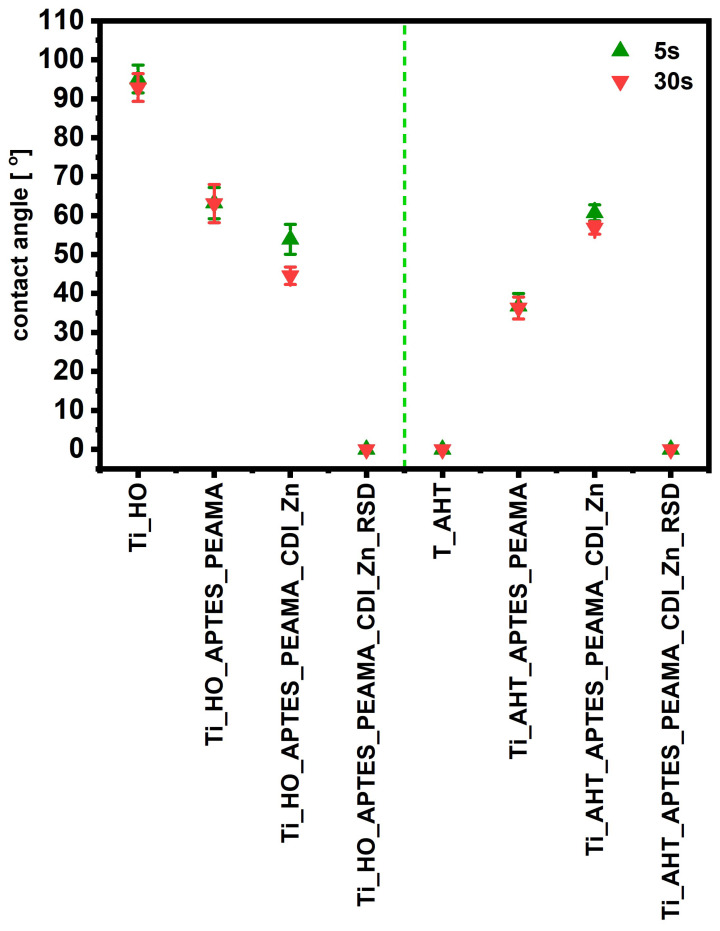
Results of the contact angles measurement after 5 s and 30 s of applying the water drops.

**Figure 10 materials-16-05404-f010:**
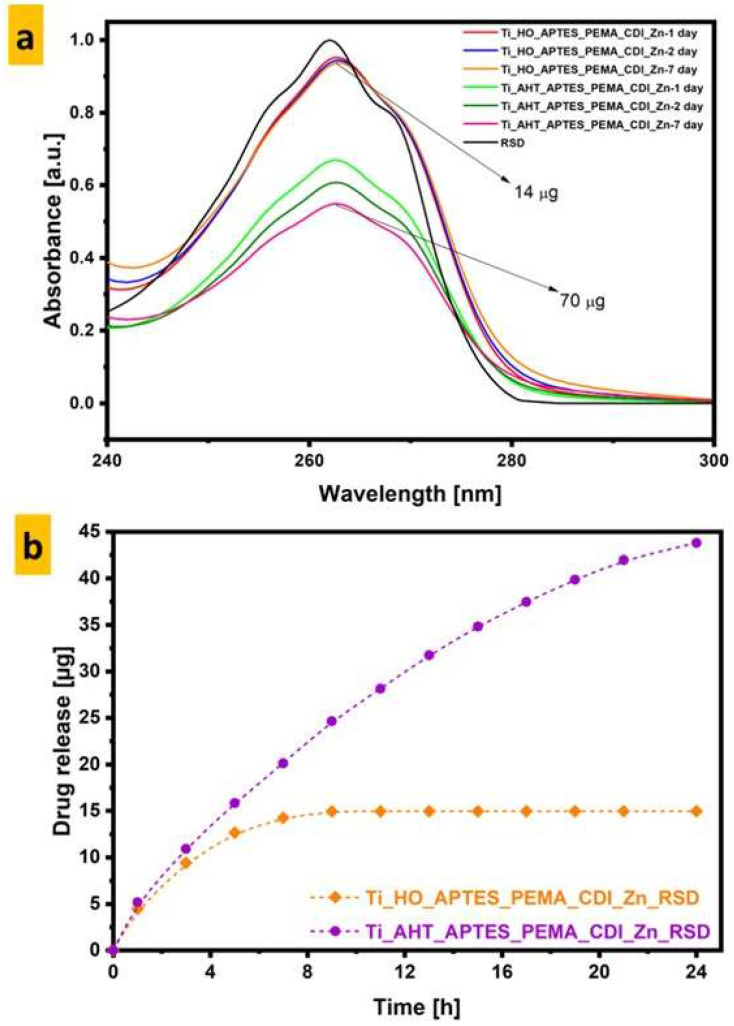
UV-Vis spectra of: (**a**)—sorption of RSD after 1st, 2nd, and 7th days; (**b**)—desorption of RSD during 24 h.

**Figure 11 materials-16-05404-f011:**
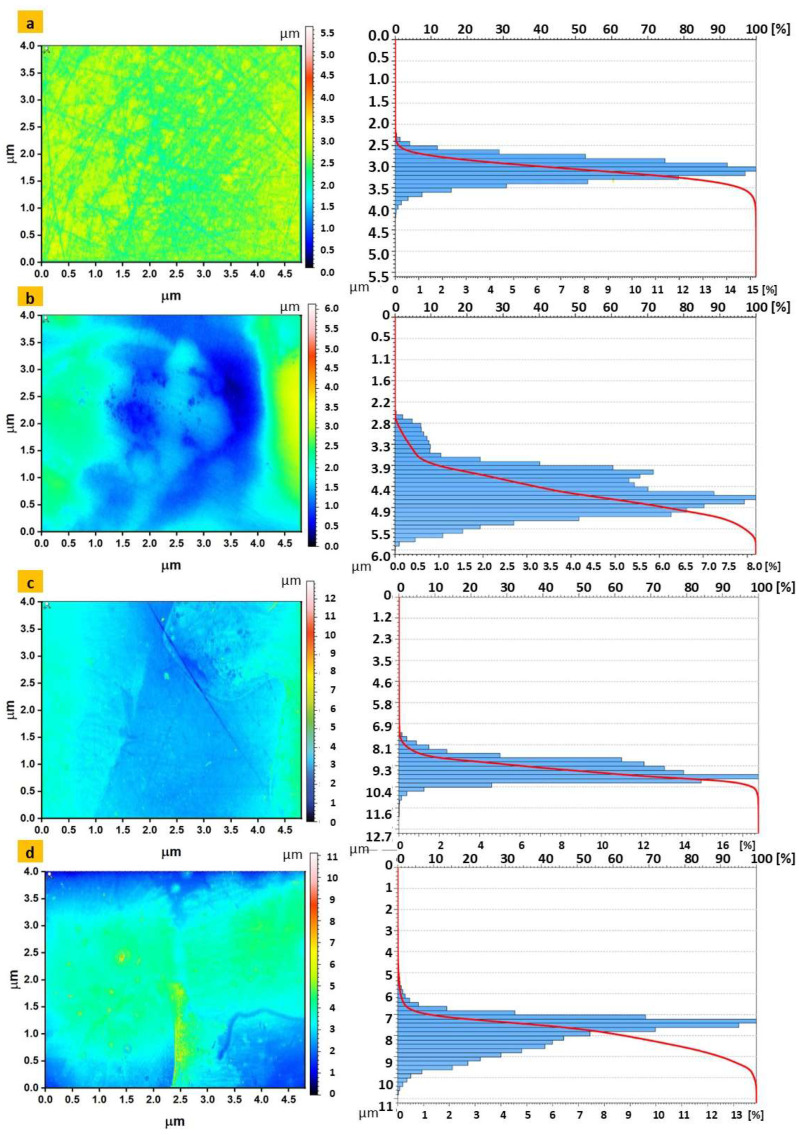
Measurement results for the surface topography (2D) and the obtained Abbott curves for the samples: (**a**)—Ti_HO; (**b**)—Ti_HO_APTES_PEAMA_CDI_Zn_RSD; (**c**)—Ti_AHT; (**d**)—Ti_AHT_APTES_PEAMA_CDI_Zn_RSD.

**Table 1 materials-16-05404-t001:** Brief and full description of prepared samples.

Brief Description	Full Description
Ti_HO	titanium alloy plate after immersion in H_2_O_2_
Ti_HO_APTES_PEAMA	titanium alloy plate after immersion in H_2_O_2_, silanization process (APTES), and polymer attachment (PEAMA)
Ti_HO_APTES_PEAMA_CDI_Zn	titanium alloy plate after H_2_O_2_ process, silanization process (APTES), polymer attachment (PEAMA), and Zn complexing by CDI
Ti_HO_APTES_PEAMA_CDI_Zn_RSD	titanium alloy plate after immersion in H_2_O_2_, silanization process (APTES), polymer attachment (PEAMA), Zn complexing by CDI, and risedronate attachment
Ti_AHT	titanium alloy plate after AHT process
Ti_AHT_APTES_PEAMA	titanium alloy plate after immersion in AHT, silanization process (APTES), and polymer attachment (PEAMA)
Ti_AHT_APTES_PEAMA_CDI_Zn	titanium alloy plate after AHT process, silanization process (APTES), polymer attachment (PEAMA), and Zn complexing by CDI
Ti_AHT_APTES_PEAMA_CDI_Zn_RSD	titanium alloy plate after AHT process, silanisation process (APTES), polymer attachment (PEAMA), exchange of Zn ions by CDI and attachment of risedronate

**Table 2 materials-16-05404-t002:** Selected topographic parameters—Rt, Rg, and Sa for the samples initial and final modification.

Name of the Studiable	R_t_		R_q_		S_a_
Sample	Mean [µm]	Std. Dev [µm]	Mean [µm]	Std. Dev [µm]	[µm]
Ti_HO	1.798	0.479	0.207	0.013	0.204
Ti_HO_APTES_PEAMA_CDI_Zn_RSD	0.942	0.270	0.103	0.018	0.475
Ti_AHT	2.836	1.200	0.201	0.044	0.453
Ti_AHT_APTES_PEAMA_CDI_Zn_RSD	3.152	1.383	0.230	0.075	0.639

## Data Availability

Not applicable.

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
