# Peer review of "Surface Modification of Ti6Al4V ELI Titanium Alloy by Poly(ethylene-alt-maleic anhydride) and Risedronate Sodium"

_materials, 2023, doi:10.3390/ma16155404_

Round 1
Reviewer 1 Report
Full Title: Surface modification of Ti6Al4V ELI titanium alloy by 2 poly(ethylene-alt-maleic anhydride) and risedronate sodium.
The above mentioned manuscript, A polymer layer of poly(ethylene-alt-maleic anhy- dride) (PEAMA) was obtained, to which the drug risedronate sodium (RSD) (an active substance in osteoporosis medication) was attached. The obtained layers were analyzed using Raman spectroscopy (spectra and maps), FT-IR ATR (spectra and maps), contact angle meas- urements as well as SEM and EDS imaging. The abstract not covers all the research work mentioned in paper. The introduction is need to be more comprehensive and the experimental work is not discussed enough. Some comments could be summarized as follows:
1- In the abstract section, : With the simultaneous increase in the number of endoprostheses being performed, ad- vances in the field of biomaterials are becoming apparent - whereby the materials and technologies used to construct implants clearly improve the quality of the implants and, ultimately, the life of the patient after surgery. ? maybe is better to moved and discussed in introduction with some details
2- In the abstract section, the authors should report their data with the corresponding reproducibility
3- Refer in the introduction part, what is the reason for choosing your samples in this research?
4- Some suffix mistakes in the paper should be corrected? The paper is seen as draft, it need to be revised?
5- labeled of y axis in Figure 7 of FTIR is not mentioned
6- The objectives are not clear – need to mention the objective of the work in the last paragraph of the introduction part?
7- Figure 9: could authors tell me how they estimate the results uncertainty of contact angle?
8- The introduction is need to be more comprehensive? The cited references is not enough? May be you can use the reference in this field such as
· Enhanced dielectric properties of flexible Cu/polymer nanocomposite films
9- Why the surface roughness not measured in paper, which is the main impactor?
10- Some details of the materials used such as the weight ratios, the chemicals used, should be mentioned in the experimental section.
11- How to measure the sample thickness?
12- In the FTIR and Ramman part, the authors should give the reasons why peak is changed?
13- The part of UV-VIS spectroscopy in the result section need to be explained enough?
14- Conclusions should to be more directed toward applied these samples in different applications?
15- Results should be described with some comparative with other works?
16- You must to increase and update the refs, most refs are old? For example ref. No 1 is not completed?
The paper accepted after the incorporation of above corrections

Reviewer 2 Report
The research article entitled: Surface modification of Ti6Al4V ELI titanium alloy by poly(ethylene-alt-maleic anhydride) and risedronate sodium has some potential for application, but is not suitable for publication. The presentation of this study is not on a good level. Nevertheless, a major revision is necessary to improve the quality of this manuscript before publication. The following issues should be addressed:
1. The abstract should provide the intention/purpose/need for this study and a suitable application for the material. This abstract deals only with investigation methods and results, which is not enough. Moreover, abbreviations should be avoided in the abstract (or at least defined) for a better readability and only main results that support the working thesis should be mentioned.
2. The keywords are not separated by a semicolon.
3. The introduction is very short and lacking of other methods that are suitable to surface modify of Ti alloys.
4. Figure 1: No, it is not an “Illustrative diagram of the research”! Where are the overview of the utilized samples? Where are the utilized investigation methods? This figure deals only with the applied surface modification process. Moreover, the drawing is too small, the labeling are too small and difficult to read. At minimum, the sample substrate should indicate correctly. Space signs are missing in the text fields of this figure.
4. The used chemicals should be mentioned correctly: (purity/molecular weight for polymers, manufacturer, city of manufacturing, country of manufacturing).
5. Throughout the whole manuscript, in figures and in the manuscript text between a value and a unit space signs are missing.
6. The utilized devices are not always mentioned correctly: (type, company, city, country).
7. Figure 2: Scale bars and labeling are too small. Moreover, space sings are missing in the scale bars.
8. Figure 3, 4 and 10: This figure is too small! All is not good readable.
9. According to figure 2, the roughness of the roughness of the samples should be determined and discussed.
10. The surface energy should be determined, since this surface modification is for implants.
11. The surface modification is with zinc. It is necessary to determine the zinc release in body simulating liquids over time; at least for a week. At this time, also the changes in pH should be recorded, since big changes can indicated a cytotoxic properties for the coatings. Is the released amount of zinc over time cytotoxic?
12. Since zinc compounds are used in this study, the antibacterial properties should be also indicated, since this is of a high interest in implant coatings.
13. The discussion of the results is poor and should be expanded.
Space signs should be very carefully checked, since this is a big issue in this manuscript.
Round 2
Reviewer 2 Report
The authors of the research article entitled Surface modification of Ti6Al4V ELI titanium alloy by poly(ethylene-alt-maleic anhydride) and risedronate sodium improved the manuscript significantly. Some points are still not satifying, but the work can be recommended for publication.